# High Dynamic Range Modulo Imaging
# for Robust Object Detection
# in Autonomous Driving

Kebin Contreras[2] , Brayan Monroy[1], and Jorge Bacca[1]*

[1] Universidad Industrial de Santander, Bucaramanga, Colombia
[2] Universidad del Cauca, Popayán, Colombia

**Abstract.** Object detection precision is crucial for ensuring the safety and efficacy of autonomous driving systems. The quality of acquired images directly influences the ability of autonomous driving systems to correctly recognize and respond to other vehicles, pedestrians, and obstacles in real-time. However, real environments present extreme variations in lighting, causing saturation problems and resulting in the loss of crucial details for detection. Traditionally, High Dynamic Range (HDR) images have been preferred for their ability to capture a broad spectrum of light intensities, but the need for multiple captures to construct HDR images is inefficient for real-time applications in autonomous vehicles. To address these issues, this work introduces the use of modulo sensors for robust object detection. The modulo sensor allows pixels to 'reset/wrap' upon reaching saturation level by acquiring an irradiance encoding image which can then be recovered using unwrapping algorithms. The applied reconstruction techniques enable HDR recovery of color intensity and image details, ensuring better visual quality even under extreme lighting conditions at the cost of extra time. Experiments with the YOLOv10 model demonstrate that images processed using modulo images achieve performance comparable to HDR images and significantly surpass saturated images in terms of object detection accuracy. Moreover, the proposed modulo imaging step combined with HDR image reconstruction is shorter than the time required for conventional HDR image acquisition.

**Keywords:** Autonomous Driving · Object Detection · Modulo Imaging.

## 1 Introduction

Object detection is a critical component of autonomous driving. The ability to accurately identify and respond appropriately to other vehicles, pedestrians, and obstacles depends largely on the quality of the acquired images. High-quality imaging is vital for enabling precise object detection, which is essential for the safe operation of autonomous vehicles in both urban and highway environments [16]. One of the acquisition problems is saturation, which can be

* This work was supported by VIE-UIS, under project 3968 and 3924.

identified as areas where pixels have reached maximum intensity, losing important information about the scene [6]. Addressing light saturation issues is crucial for the accuracy of object detection systems. Figure 1 illustrates common challenges such as overexposure due to direct sunlight (a), reflections of artificial nightlights (b) and loss of road details due to solar saturation (c), encountered in autonomous driving environments. An alternative is the well-known High Dynamic Range (HDR) imaging technique, which combines multiple exposure images to capture a broader spectrum of light intensities [15]. This approach effectively addresses the issues of underexposure and overexposure. However, the time required to process these images makes them impractical for real-time applications in autonomous driving [9].

A solution to address these challenges is the introduction of modulo sensor technology, which employs an irradiance encoding scheme. This technology allows pixels to 'reset/wrap' upon reaching saturation, thereby enabling continuous data recording. Although modulo images do not directly maintain the integrity of color at the moment of capture, they preserve sufficient details to enhance object detection. Furthermore, due to similarities with phase unwrapping problems, certain unwrapping algorithms can be employed to obtain HDR images during post-processing, which is crucial for achieving high visual quality under variable lighting conditions [11,19]. This technology is especially valuable in dynamic environments where lighting conditions can change rapidly and dramatically, providing a solution to preserve image information in cluttered areas.

To the best of our knowledge, this is the first work to introduce the modulo sensor as an alternative for obtaining HDR images to enhance robust object detection in autonomous driving. We specifically evaluate the performance of object detection using YOLOv10 [14] models in various saturation scenarios and demonstrate that modulo measurements outperform commercial images under extreme conditions without the need for retraining. Furthermore, we propose to adapt the Simultaneous Phase Unwrapping and Denoising (SPUD) algorithm for direct HDR estimation from modulo images [10]. This approach significantly improves object detection results without significantly increasing processing time, making the modulo sensor a promising solution for autonomous driving systems.

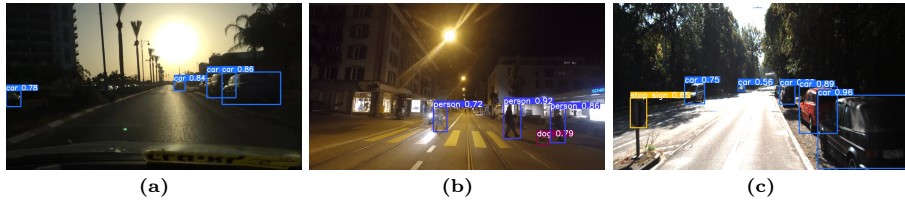

(a)          (b)          (c)

**Fig. 1:** Challenges of sensor saturation in autonomous driving environments from various databases: (a) BDD100K shows problems of overexposure due to direct sunlight [17]; (b) Nighttime Driving depicts reflections from artificial night lights [4]; and (c) KITTI demonstrates the loss of road details due to solar saturation [5].

## 2    Background

**Modulo Imaging.** The image formation process in a modulo sensor involves the use of a modulo operator to reset the intensity values upon reaching a specific threshold of $2^b$. This imaging process can be mathematically modeled as follows

$$y = Q_b(\mathrm{mod}(x, 2^b)) = W_b(x), \tag{1}$$

where $y$ represents the acquired modulo image, $Q_b(\cdot)$ represents the quantization process to $b$ bits, $x$ denotes the desired HDR image and $\mathrm{mod}(\cdot)$ refers to the modulo operator. The implementation of this particular imaging process has been explored with self-reset Analog-to-Digital Converter (ADC) [18] and with programmable sensors [11]. More recently, with the advancements of the Unlimited Sampling Framework, it is now possible to implement it using modulo-ADCs in continuous time, i.e., prior sampling [2]. Once the modulo image $y$ is captured, a recovery process is needed to get an estimation of the HDR image $x$. However, this recovery process is an optional step since the object detection network can be employed directly to raw modulo image.

**Modulo Recovery.** Given that the modulo operator is a non-linear transformation, it is impractical to directly employ numerical optimization techniques to reconstruct from the modulo image $y$. However, similar to the unwrapping problem, the spatial differences in the HDR image, $\Delta x$, matches with the *wrapped* spatial differences of the modulo image, expressed as $\Delta x = M_b(\Delta y)$, where $M_b(\Delta y) = W_b(\Delta y + 2^{b-1}) - 2^{b-1}$ is a centered modulo operator to align negative values [7,12]. This assumption holds if the natural image discontinuities do not exceed the threshold $2^{b-1}$, namely, $\|\Delta x\|_\infty < 2^{b-1}$, also referred to as Itoh's condition [7]. Consequently, the HDR image recovery can be mathematically formulated as the following optimization problem

$$\hat{x} \in \arg\min_x \|\Delta x - M_b(\Delta y)\|_2^2 + \tau R(x), \tag{2}$$

where the regularization operator $R(\cdot)$ is introduced to exploit additional knowledge about the structure of the HDR image $x$ or to handle challenging recovery scenarios due to the presence of noise [1,3,13]. In the subsequent section, we introduce SPUD [10], a non-interactive method to solve Equation 2 under sparsity assumption.

## 3    Methodology

In this section, we detail our proposed methodology consisting of a two-step pipeline to process traffic images using modulo sensors: (A) Acquisition and Recovery with modulo sensors and (B) Object Detection on modulo images or over HDR recovered images.

**Step A. Acquisition and Recovery with Modulo Sensors:** The first step consists of replacing the conventional CCD sensor with a modulo sensor to acquire a modulo image $y$ following Equation (1). Then, the acquired modulo image $y$ can be fed into a modulo recovery algorithm to restore the HDR image $x$. We employ SPUD [10] which computes the HDR image recovery as follows

$$\boldsymbol{\rho} = \mathcal{D}(\Delta^\top M_b(\Delta y)),$$

$$\hat{x}_{mn+n} = \mathcal{D}^{-1}\left(\frac{\mathcal{T}(\rho, \tau)_{mn+n}}{2(2 - \cos(\pi m/M) - \cos(\pi n/N))}\right). \tag{3}$$

In (3), $M$ and $N$ represent the dimensions of the image, $\mathcal{D}$ denotes the discrete cosine transform (DCT) [10], and $\mathcal{T}$ is the hard-thresholding operator with threshold parameter $\tau$ [10]. For simplicity, we denote the computation of Equation (3) by the reconstruction operator $\hat{x} = \text{SPUD}(y, b, \tau)$, effectively combining noise reduction and image unwrapping in a single step.

Although new algorithms have been used to obtain HDR images from modulo images [1,19], the effectivity of the SPUD is mainly attributed to its non-iterative approach with an order of complexity of $\mathcal{O}(n \log(n))$. This method significantly improves the quality of the restored HDR image with a low latency time, which is crucial for real-time applications such as autonomous driving [10].

**Step B. Object Detection on Modulo and Recovered Images:** The second step involves performing an autonomous driving task on the modulo image and recovered HDR images. Consequently, a family of YOLOv10 detection models is used to identify the classes present in the autonomous driving datasets during the object detection task. This process can be represented as

$$\mathcal{C} = \text{YOLO}(\hat{x}, y, p), \tag{4}$$

where $\mathcal{C} = \{\mathbf{c}^{(i)}\}_i^N$ is the set of boundary boxes provided by the YOLOv10 detection model for $N$ different objects detected in the input image, and $p$ is a boolean parameter indicating whether to use the restored HDR image $\hat{x}$ or the raw modulo image $y$. Here, each boundary box $\mathbf{c}$ is constructed as $\mathbf{c} = [j_1, k_1, j_2, k_2, z]$, where $(j_1, k_1)$ and $(j_2, k_2)$ are the top-left and bottom-right coordinates of the box, and $z$ denotes the predicted object class. We evaluate object detection using various configurations of the YOLOv10 model, including the variants $n, s, m, b, l, x$, following an ascending order in terms of complexity and the number of model parameters. The proposed two-step methodology is summarized in Algorithm 1.

---

**Algorithm 1** Object Detection with Modulo Images (ODMI)

---

1: **Input:** HDR image $x$, boolean $p$, color depth $b$, optimization parameter $\tau$
2: $y \leftarrow W_b(x)$                    ▷ Simulation of modulo sensor Equation (1)
3: $\hat{x} \leftarrow \text{SPUD}(y, b, \tau)$           ▷ HDR image reconstruction Equation (3)
4: $\mathcal{C} \leftarrow \text{YOLO}(\hat{x}, y, p)$         ▷ Calculation of boundary boxes Equation (4)
5: **Return:** $\mathcal{C}$                          ▷ Set of boundary boxes

---

## 4    Simulation and Results

This section evaluates the robustness of ODMI compared to standard digital cameras. Our main goal is to analyze the object detection performance under varying saturation conditions for different versions of the YOLOv10 detection model without retraining. For this, we emulated the acquisition of modulo images, the reconstruction of the HDR image from modulo images using the SPUD algorithm [10], and the acquisition of a saturated image in the KITTI database. Specifically, to simulate the different saturation conditions, we normalize the HDR image $x_{\text{raw}}$ to the range $[0, 2^b]$, with $b = 8$, and introduce a saturation factor $\alpha$, which scales the HDR image to $x = \alpha x_{\text{raw}}$. The saturation scenario for a standard camera is modeled as

$$x_{\text{sat}} = \min(x, 2^b), \tag{5}$$

where $x_{\text{sat}}$ is the saturated image, and $\alpha$ is the saturation factor that adjusts the image intensity to simulate different levels of light exposure. We evaluate the performance of different imaging methodologies from low to extreme saturation scenarios, by selected $\alpha$ values of 1.5, 2, 3, and 4. For emulating the modulo camera and HDR image reconstruction, we use Equation (1) and Equation (3), respectively, using the same scaled HDR image $x$ as input.

**Dataset.** The proposed methodology was evaluated using the KITTI dataset [5], which is a benchmark in object detection focused on autonomous driving. This dataset contains 7,481 images, each with dimensions of $1242 \times 375$, and captures urban scenes featuring a variety of objects, including vehicles, pedestrians, cyclists, miscellaneous objects, sitting people, trams, trucks, and vans, making it a significant challenge for vision-based object detection systems.

**Model.** The YOLOv10 model family, with configurations $n, s, m, b, l,$ and $x$ between 2.3M to 29.5M parameters, was used to assess object detection effectiveness. These models consist of a CNN backbone with detection layers and anchor boxes for predicting object-bound boxes. Additionally, the integration of partial self-attention modulo enhances feature extraction and localization accuracy [14].

**Metrics.** The object detection performance for all scenarios (image-saturated, modulo sensor, recovery, and ideal HDR) was evaluated using: Intersection over Union (IoU), the F1-score, and Accuracy. These metrics are standard benchmarks that enable an objective comparison of the performance of different YOLO-v10 variants under diverse image manipulation conditions. In the experiment, 100 testing images from the KITTI database were selected to represent different saturation conditions. The experiments were conducted on a computer equipped with an AMD Ryzen 7 5700X 8-Core Processor running at 3.40 GHz, 64.0 GB of RAM, and an NVIDIA GeForce RTX 4070 GPU with 12 GB of VRAM.

The analysis of the different image techniques under varying levels of light saturation is presented in Table 1. The models evaluated, from best to worst performance, are $x, l, b, m, s, n$, in relation to the number of trainable parameters described in Table 2. As the value of $\alpha$ increases, the IOU, F1-Score, and accuracy metrics decrease due to the loss of information in saturated areas, as shown in

**Table 1:** Quantitative Evaluation of Saturated, Modulo and Recovery imaging techniques with four saturation levels ($\alpha$ 1.5, 2, 3, 4) The best result is in bold and the second is underlined.

| Model YOLOv10 | Method | IOU ($\alpha$) | | | | F1-Score ($\alpha$) | | | | Accuracy ($\alpha$) | | | |
|---|---|---|---|---|---|---|---|---|---|---|---|---|---|
| | | 1.5 | 2 | 3 | 4 | 1.5 | 2 | 3 | 4 | 1.5 | 2 | 3 | 4 |
| **n** | Saturated | 58.9 | 55.3 | 53.4 | 50.6 | 37.8 | 37.0 | 35.2 | 35.2 | 29.8 | 28.5 | 28.5 | 25.0 |
| | Modulo | 60.2 | 58.4 | 56.2 | **53.2** | 40.0 | **40.1** | 38.6 | 36.7 | 29.8 | **29.0** | **29.0** | **28.4** |
| | Recovery | **61.0** | **59.3** | **57.4** | 53.0 | **41.1** | 40.0 | **39.7** | **39.7** | **30.2** | 28.7 | 26.8 | 26.8 |
| | Ideal HDR | | 62.1 | | | | 42.0 | | | | 32.0 | | |
| **s** | Saturated | 56.6 | 55.1 | 44.4 | 34.5 | 36.5 | 34.3 | 23.6 | 16.6 | 27.3 | 25.6 | 16.3 | 11.5 |
| | Modulo | 61.8 | 60.1 | 57.6 | **54.3** | **42.2** | 39.3 | 36.0 | 31.1 | 31.5 | 29.6 | 26.7 | 23.0 |
| | Recovery | **62.0** | **61.8** | **60.1** | 54.2 | 41.6 | **41.3** | **39.3** | **35.1** | **31.6** | **32.0** | **29.6** | **26.0** |
| | Ideal HDR | | 62.6 | | | | 42.1 | | | | 32.1 | | |
| **m** | Saturated | 61.8 | 60.7 | 58.7 | 56.8 | 40.8 | 42.3 | 40.5 | 40.5 | 38.7 | 38.1 | 37.7 | 36.1 |
| | Modulo | **65.4** | **64.3** | **64.0** | **63.6** | 43.5 | 43.0 | 42.6 | 42.7 | 40.0 | 38.7 | 36.7 | 35.1 |
| | Recovery | **65.4** | 64.2 | 63.5 | 63.5 | **44.2** | **43.6** | **43.5** | **43.1** | **41.1** | **40.0** | **39.8** | **39.0** |
| | Ideal HDR | | 65.8 | | | | 45.9 | | | | 41.2 | | |
| **b** | Saturated | 65.4 | 65.0 | 64.8 | 64.7 | 43.5 | 43.2 | 42.6 | 41.8 | 39.8 | 38.7 | 37.9 | 35.7 |
| | Modulo | 67.3 | 67.1 | 66.7 | 66.4 | 44.4 | 43.6 | 42.6 | 41.8 | 41.1 | 40.7 | 39.6 | 38.6 |
| | Recovery | **68.3** | **68.0** | **67.3** | **67.0** | **49.3** | **49.0** | **48.3** | **48.1** | **42.2** | **41.2** | **40.3** | **40.1** |
| | Ideal HDR | | 70.0 | | | | 51.2 | | | | 43.6 | | |
| **l** | Saturated | 65.1 | 63.0 | 57.1 | 47.8 | 44.6 | 42.3 | 33.0 | 24.8 | 34.3 | 32.1 | 23.3 | 17.6 |
| | Modulo | 69.4 | 68.7 | 65.4 | 58.8 | **53.3** | **53.3** | 46.2 | 38.4 | 41.6 | 40.9 | 36.0 | 28.5 |
| | Recovery | **69.9** | **69.9** | **68.4** | **60.6** | 52.7 | 52.5 | **50.3** | **42.2** | **42.0** | **41.8** | **36.1** | **31.9** |
| | Ideal HDR | | 71.3 | | | | 56.3 | | | | 45.5 | | |
| **x** | Saturated | 65.5 | 63.1 | 57.8 | 48.8 | 44.9 | 42.2 | 35.4 | 27.1 | 34.3 | 31.4 | 25.9 | 19.3 |
| | Modulo | 71.0 | 70.0 | 67.9 | 62.3 | 54.8 | 54.1 | 51.7 | 43.9 | 44.1 | 42.4 | 41.2 | 33.6 |
| | Recovery | **72.0** | **71.2** | **70.7** | **67.9** | **56.1** | **55.7** | **52.1** | **48.1** | **45.2** | **45.1** | **42.2** | **36.2** |
| | Ideal HDR | | 72.9 | | | | 62.3 | | | | 50.2 | | |

Figure 2. Specifically, for $\alpha = 3$, it is possible to detect a person in HDR, Modulo, and Recovery images, whereas detection is not possible in the Saturated image. For $\alpha = 4$ and $\alpha = 5$, the detection model fails to detect objects in the Saturated image, while Modulo and Recovery images still allow object detection, as reflected in the numerical results in Table 1. Notice that the metric in Table 1 for the ideal HDR image is constant for different $\alpha$ values because the HDR image is not affected by $\alpha$ and does not suffer from saturation since we assume perfect recovery of the HDR image. This analysis shows that Modulo and Recovery image offer better spatial information and robust object detection under high saturation compared to the Saturated image. The Ideal HDR image, unaffected by saturation, serves as a reference for evaluation as it represents the ideal scene.

**Fig. 2:** Comparison of imaging techniques under varying levels of light saturation. Each column stands for; Ideal HDR, Saturated, Modulo, and Recovery. Each row shows the results for different values of the saturation factor $\alpha$, ranging from 1.5 to 4.

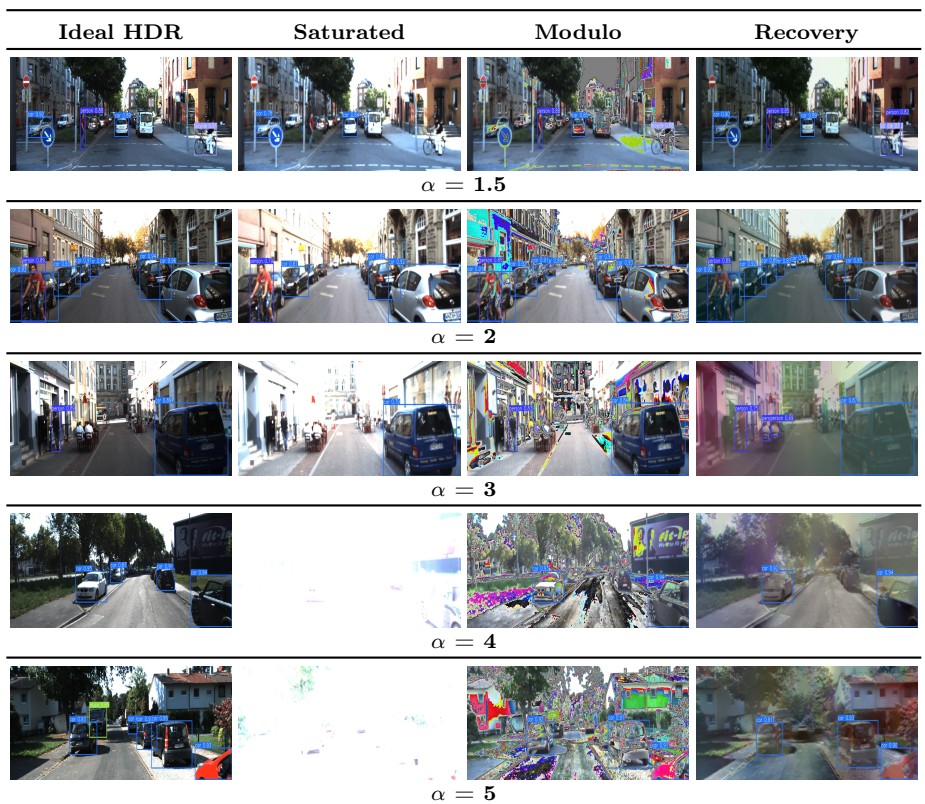

By simulating the processes of saturation, modulo sensor, reconstruction, and ideal HDR, we can compare the inference times between different methodologies. For the Saturated and Modulo cases, we assume an ideal camera with an acquisition time of 33 ms, incorporating the inference time of the YOLOv10 models. For the Recovery case, the inference time includes the SPUD algorithm time of 5.1 ms $\pm$ 1.9 ms, using different of $\alpha$ value (1.5, 2, 3, 4). For ideal HDR images, we assume three captures of 33 ms each to obtain the final HDR image. This approach follows the standard procedure of using three images taken at different exposure times to create an HDR image presented in [8]. Table 2 summarizes the parameters and inference times for each model. As YOLO complexity decreases, the parameters and inference times is reduced. Recovery time from a modulo image is generally lower than the time for ideal HDR images using the conventional multiple exposure time technique. The Recovery time encompasses both image reconstruction and YOLO detection, whereas the remaining time refers exclusively to detection.

**Table 2:** Comparison of computational time employing YOLOv10 models.

| Model YOLOV10 | No. Parameters [M] | Time [ms] | | |
|---|---|---|---|---|
| | | Saturated/Modulo | Recovery | Ideal HDR |
| n | 2.3 M | 56.18 ± 0.2 | 62.95 ± 0.3 | 122.85 ± 0.1 |
| s | 7.2 M | 57.23 ± 0.1 | 65.34 ± 0.4 | 123.97± 0.2 |
| m | 15.4 M | 59.32 ± 0.2 | 68.23 ± 0.5 | 126.17 ± 0.1 |
| b | 19.1 M | 59.87 ± 0.3 | 71.34 ± 0.3 | 128.17± 0.3 |
| l | 24.4 M | 63.56 ± 0.3 | 73.49 ± 0.3 | 131.68 ± 0.4 |
| x | 29.5 M | 65.84 ± 0.7 | 79.25 ± 0.5 | 136.34 ± 0.5 |

## 5   Conclusion

This work proposes the use of modulo sensor combined with the SPUD reconstruction algorithm to enhance object detection in autonomous driving under high illumination conditions. Our results show a significant improvement over standard cameras, which are often affected by saturation issues. SPUD reconstructed images closely approximate HDR quality for object detection, demonstrating that phase unwrapping with modulo sensors achieves comparable results. Additionally, HDR recovery from modulo images is processed faster than multi-exposure HDR images, which benefits real-time processing in dynamic environments. The most complex version of the YOLOv10 model $x$) performed best, followed by versions $l, b, m, s$, and $n$, highlighting the advantage of increased model complexity for dynamic imaging. Interestingly, the object detection quality of the saturated images decreases significantly compared to the modulo sensor, confirming the effectiveness of this new sensor in preserving detail and maintaining higher object detection accuracy.

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
