# Supplementary Material

This section evaluates some visual results using the same scene.

**Fig. 1:** Comparison of different imaging techniques under varying levels of light saturation. The columns represent three different methods: Saturated, Modulo, and Recovery. The rows illustrate the results for different values of the scaling factor $\alpha$, ranging from 1.5 to 8.

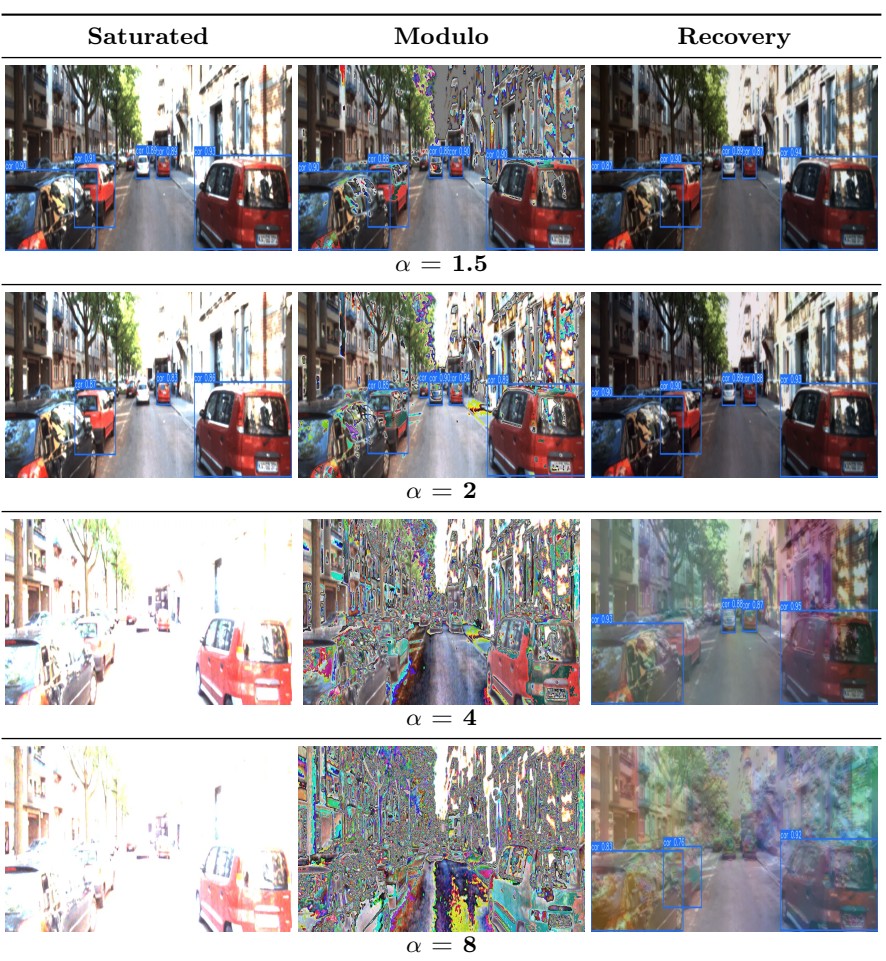