# OpenReview forum: "High Dynamic Range Modulo Imaging for Robust Object Detection in Autonomous Driving"
_thecvf.com/ECCV/2024/Workshop/ROAM — ROAM ECCV 2024 Oral_

### Official Review · Reviewer_vieB · 2024-08-11
**Modulo imaging for handling extreme conditions**

**Rating:** 8
**Confidence:** 4

**Review:**

**Summary and Contributions**

The paper proposes the use of modulo sensors for enabling object detection in extreme lighting conditions. The authors utilize modulo imaging and the SPUD algorithm to recover an estimated HDR image in O(n log(n)) time. This way, the quality of the restored HDR image makes it possible to use conventional object detection methods that otherwise would fail on the saturated images.

The paper is well-written and easy to follow. The motivation is clear and the problem is important since saturation can severely decrease detection performance and hinder downstream tasks built on the perception stack. The experimental results are promising and seem to support the author's claims. Furthermore, the runtime analysis helps to evaluate whether the method can be further used in real time.

**Strengths**
- The problem addressed by the paper is important and relevant to the workshop.
- Introducing modulo sensing as an alternative for obtaining HDR images is a good idea.
- Well-designed experiments with promising initial results.

**Opportunities for Improvement**
- Increase the test set size and use other datasets for investigating the generalization ability of the proposed method
- It would be great to see how the proposed method works on real-world data instead of the emulated one. This would require the collection of a dataset which is a large effort but would aid the evaluation of the method.
- Extending the experiments with different models (e.g., segmentation, depth) would increase the value of the paper.

---

### Official Review · Reviewer_xZe7 · 2024-08-16
**Review: High Dynamic Range Modulo Imaging for Robust Object Detection in Autonomous Driving**

**Rating:** 7
**Confidence:** 3

**Review:**

Summary:
To address the problem of object detection in extreme lighting conditions, this paper proposes the novel usage of modulo sensors and adaption of Simultaneous Phase Unwrapping and Denoising (SPUD) algorithm. Such approaches to recover HDR images from modulo images improves results greatly with a reasonable impact to efficiency. The authors also evaluate their approach using YOLOv10 and confirm the effectiveness.

Pros:
- Novel idea
- Convincing and promising preliminary results
- Clear structure and easy to follow

Cons:
- Could have more comparisons with prior methods
- Limited model/dataset (sufficient for a workshop paper, but definitely benefits to have more)

---

### Official Review · Reviewer_v4NV · 2024-08-16
**Simulated modulo camera performs well on downstream object-detection task**

**Rating:** 7
**Confidence:** 4

**Review:**

Summary and Contributions

The Authors evaluated the zero-shot object-detection performance of pretrained YOLOv10 models on simulated modulo camera images created from the KITTI benchmark images using manual overexposure. The object-detection benchmark results were compared on raw overexposed images, simulated modulo images and HDR images recovered from the modulo images using the real-time SPUD algorithm. The Authors successfully demonstrate a significant performance increase by running the models on modulo images against the raw oversaturated images. Moreover, HDR recovery has been shown to further increase the performance for most of the models and overexposure rates at a minor computational cost.

Strengths

Saturation of images is a general limitation of conventional CCD cameras in highly dynamic scenes, which is especially relevant in autonomous driving. Sudden changes in brightness can cause overexposure, causing serious degradation of camera-based perception. This work examines whether using modulo-camera images along with a subsequent phase-unwrapping recovery algorithm could restore the quality of downstream perception tasks, by retrieving details lost by conventional cameras. As a generic perception task, object detection has been evaluated. The reported results are quite convincing, suggesting that using modulo cameras could be a viable choice to mitigate the degradation caued by overexposure.

Opportunities for Improvement

Whereas object detection is indeed a crucial part of the perception stack of most autonomous driving systems, other tasks should be evaluated as well. Especially dense pixel-level tasks such as segmentation or depth estimation could be affected by the false colors introduced by the modulo sampling or a noisy recovery. Nevertheless, the recovered images apparently exhibit much more details than the saturated images, which probably aids any downstream computer-vision task. In this manner the reported results confirm this naiive expectation.
On the other hand, the domain gap between the recovered modulo images and the original unsaturated HDR images the models have been trained on seems to be still significant. A possible follow-up work could assess the usage of different recovery algorithms to close this domain gap.
Furthermore the object-detection models could be trained on the KITTI dataset using simulated modulo images and the performance of the models could be re-assessed. This would establish a fair comparison between the performance on the modulo images and the raw HDR baseline.

Limitations

Even though modulo cameras have been proposed quite a while ago, I have not heard about them being used in autonomous driving systems. Even this work does not evaluate images obtained from such a camera, which makes me assume that there is some fundamental limitation why these hardwares are not so widespread. I am missing a more detailed discussion of strenghts and limitations of such cameras for industry use. Also, the results would have been even more convincing if real modulo-camera images had been used for evaluation instead of the simulated ones. However, I understand that it is straightforward to use a well-established benchmark and the artificially oversaturated images provide a good baseline for the comparison.
Whereas I find the paper quite intriguing, I am missing the novelty component. The potential of modulo imaging to recover details of oversaturated images has already been demonstrated and several different HDR recovery algorithms have been proposed (Zhao et al ICCP 2015, Zhou et al NIPS 2020, Pineda J. et al Applied Optics 2020). The fact that object detection is also improved is quite expected due the recovered details in the image. On the other hand there is no discussion of the reduced performance with respect to the unsaturated baseline, which would really increase its impact.

Relation To Prior Work

There is a short section titled Background, however a more extensive discussion of the pros and cons of modulo cameras is missing. Also there is no overview of the usage of such cameras in the industry. Some reconstruction algorithms are mentioned in the text, but they are not discussed extensively.

Conclusion

Altogether, I think the main virtue of the paper is popularizing modulo imaging as a possible alternative to conventional cameras. With all the missing details and lack of discussion of the possible limitations of the technique, I still think that the paper is worth publishing at the ROAM workshop, as modulo imaging may be a novelty for many researchers in the computer-vision community.

---

### Official Review · Reviewer_5daZ · 2024-08-19
**Modulo sensor for robust object detection**

**Rating:** 7
**Confidence:** 4

**Review:**

This paper proposed robust objection detection by processing through modulo image steps. Experiments are conducted using YOLO v10 model demonstrating higher object detection accuracy compared to HDR images. Overall the paper is well-motivated and clearly structured. The results shown are convincing. I have two comments that are worth the authors looking into:
 (1)  In Table 1, quantitative evaluation of saturated, modulo, and recovery imaging are conducted using accuracy metrics. I notice that YOLO v10 model "m" accuracy is higher than YOLO v10 "b". Considering YOLO v10 "b" contains more parameters, the accuracy should be higher than YOLO v10 "m". This is contradictory to other YOLO models as well as other metrics.

(2) It would be more convincing if authors could test the algorithm in more datasets not just KITTI.

---

### Official Review · Reviewer_cyPu · 2024-08-19
**HDR modulo imaging for robust object detection in autonomous driving - Review**

**Rating:** 7
**Confidence:** 4

**Review:**

The authors have provided an excerpt from a paper that discusses the use of High Dynamic Range (HDR) modulo imaging for robust object detection in autonomous driving. The paper outlines the challenges of imaging under varying lighting conditions, where traditional HDR imaging methods are too slow for real-time applications. The authors propose using modulo sensors, which reset upon reaching saturation, allowing for HDR recovery through unwrapping algorithms.

The methodology is tested on object detection using various configurations of the YOLOv10 model, demonstrating that the modulo imaging technique improves detection performance compared to traditional saturated images, especially under extreme lighting conditions. The results highlight that modulo images combined with HDR recovery can be processed more efficiently than conventional HDR acquisition methods, making this approach suitable for real-time autonomous driving scenarios.

The paper appears to be thorough, including mathematical modeling, algorithmic details, and experimental evaluations, with promising results for real-world applications in autonomous driving.


### Pros:
1. **Better Dynamic Range:** Captures bright and dark areas clearly, enhancing visibility in tough lighting.
2. **Improved Object Detection:** Enhances accuracy, especially in challenging conditions.
3. **Less Noise:** Produces clearer images with reduced graininess in low light.
4. **Flexible Design:** Easier to integrate into existing systems.
5. **Effective in Difficult Scenarios:** Performs well in tricky environments like nighttime driving.

### Cons:
1. **High Processing Demand:** Requires significant computing power, possibly slowing performance.
2. **Expensive Hardware:** May need costly sensors or components.
3. **Integration Challenges:** Potential difficulties in fitting with current vehicle systems.
4. **Risk of Visual Errors:** Could produce inaccuracies in fast-moving settings.
5. **Limited Testing:** Might not be fully proven in diverse real-world conditions.

---

### Official Review · Reviewer_AYsz · 2024-08-20
**Modulo Sensor as potential alternative of HDR imaging for object detection.**

**Rating:** 8
**Confidence:** 4

**Review:**

The authors argue that, for the purpose of object detection, full HDR image capturing can be substituted with a Modulo sensor based camera. A mainstream object detection network could use the raw image with the wrapped around intensity values or the "unwrapped" HDR image as input. For unwrapping the authors use the SPUD algorithm.
To support these claims the paper compares the performance of a popular pretrained object detection model on the emulated raw modulo image, the reconstructed HDR and the original HDR image which was used  to produce the modulo images in the first place by artificially oversaturating them.

### **Strengths**:
1. The paper is well structured and easy to follow
2. Tackles a relevant issue of information loss due to pixel saturation and runtime improvements.
3. Using both the raw and the reconstructed images for demonstration.

### **Opportunities for improvement**:
1. Expanded test set for emulated data.
2. Simulating the all too frequent local oversaturation (glare) would be welcome addition to the uniformly stretched intensity.
3. Fig 1 in the supplementary material is easier to follow than Fig 2 in the main paper as it uses the same image over the whole alpha range.
4. I understand the difficulties but acquiring real world modulo sensor images and putting them to the test would be convincing.

---

### Official Review · Reviewer_skm5 · 2024-08-20
**Module sensors enhances object detection in saturated images**

**Rating:** 8
**Confidence:** 4

**Review:**

The authors propose the use of modulo sensors for HDR imaging in autonomous driving, addressing critical issues with saturation in extreme lighting conditions. The experimental results demonstrate the effectiveness of the YOLOv10 model combined with SPUD reconstruction. The fact that modulo imaging outperforms conventional HDR methods in both detection accuracy and processing time is interesting and relevant to the workshop.


Pros:
Relevance: The method is directly applicable to autonomous vehicles, improving object detection accuracy under challenging conditions.
Performance Gains: Experimental results show significant improvements in detection accuracy and processing time compared to traditional HDR methods.

Cons:
Implementation Complexity: Incorporating modulo sensors and SPUD reconstruction adds complexity to the system, particularly it still requires the computation of the HDR image x.
Processing Overhead: The additional reconstruction step may introduce some processing delays, which could be a concern in real-time applications. The authors didn't mention how to accelerate efficient inference methods (quantization methods) to achieve near real-time object prediction while computing modulo imaging on the fly.

Recommendation:
Accept. The paper proposes a novel solution and clear experimental benefits show their benefits.

---

### Official Review · Reviewer_LPeL · 2024-08-22
**HDR modulo imaging for robust object detection in autonomous driving**

**Rating:** 8
**Confidence:** 4

**Review:**

Presents an innovative method utilizing module sensors to improve object detection in autonomous vehicles, effectively tackling the challenge of light saturation. The approach is robust, and the experimental results show that this technique can surpass saturated images and compete with traditional HDR images in both accuracy and efficiency. The paper could strengthen its argument by providing more extensive experimental results, including diverse lighting conditions and a broader set of scenarios.

---

### Decision · Program_Chairs · 2024-08-22

**Decision:**

Accept (Oral)

**Comment:**

The average score of the paper given all the reviews received before the deadline was higher than 5.5 (1 is lowest, 10 is highest), therefore the paper is accepted. The Authors are encouraged to consider feedback for the camera ready version of the paper due on August 31st.

**This was also the highest scoring paper of all submitted papers to the ROAM workshop of ECCV 2024. Congratulations to all the Authors!**